# Significant Reversal of Facial Wrinkle, Pigmented Spot and Roughness by Daily Application of *Galactomyces* Ferment Filtrate-Containing Skin Products for 12 Months—An 11-Year Longitudinal Skin Aging Rejuvenation Study

**DOI:** 10.3390/jcm12031168

**Published:** 2023-02-01

**Authors:** Kukizo Miyamoto, Yasuko Inoue, Xianghong Yan, Shiomi Yagi, Sudarsana Suda, Masutaka Furue

**Affiliations:** 1Research and Development, Kobe Innovation Center, Procter and Gamble Innovation GK, Kobe 651-0088, Japan; 2Department of Dermatology, Graduate School of Medical Sciences, Kyushu University, Fukuoka 812-8582, Japan

**Keywords:** facial skin aging, skin hydration, TEWL, hyperpigmented spot, wrinkle, texture, elasticity, pore, SK-II Pitera™ formula, *Galactomyces* ferment filtrate

## Abstract

Facial skin aging is an important psychophysical and social concern, especially in women. We compared facial parameters reflecting aging of the skin in 1999 and 2010 in 86 female volunteers. Then, all subjects applied three *Galactomyces* ferment filtrate-containing skin care products (G3 products; SK-II Facial Treatment Essence, SK-II Cellumination Essence, and SK-II Skin Signature Cream) twice daily for 12 months (M), with the skin parameters being measured at 2 M, 8 M, and 12 M during this period. Facial skin aging parameters such as wrinkles, hyperpigmented spots, and roughness significantly deteriorated during the 11-year interval. This 11-year aging process was associated with reduced hydration and increased transepidermal water loss (TEWL). Notably, treatment with G3 products significantly and cumulatively increased skin hydration with a correlated reduction of TEWL during the 12 M treatment period. Such treatment also significantly and cumulatively reversed the 11-year facial skin aging in the three parameters of wrinkles, spots, and roughness. These results suggest that facial skin retains the potential to recover from the aging process when it is applied with appropriate cosmetic agents.

## 1. Introduction

Facial skin aging is a psychophysical and social concern in humans, especially women. Facial skin aging involves changes of the skin with increasing age such as wrinkles, hyperpigmented spots, roughness, and loss of elasticity [1,2,3,4]. Wrinkles are known to increase more rapidly and more conspicuously in Caucasian than in Chinese women. Meanwhile, hyperpigmented spots are more apparent in Chinese than in Caucasian women [4]. Skin barrier function also tends to be impaired by facial skin aging, as assessed by decreased skin hydration and increased transepidermal water loss (TEWL) [1].

The intensity and speed of the aging process differ markedly between individuals [1]. Therefore, some women look younger than their actual age, while others look much older [1]. Among various skin aging parameters, wrinkles, hyperpigmented spots, and skin roughness are considered to be particularly representative of an older appearance of the female face [1]. A previous study also indicated the possibility that maintaining good facial moisturization may contribute to a younger-looking facial appearance [1]. However, it remains unknown whether regular and daily skin care with an efficient moisturizer can reverse the long-term aging process.

SK-II Pitera™ contains *Galactomyces* ferment filtrate (GFF), which works as a potent antioxidative agonist for aryl hydrocarbon receptor [5,6]. GFF is known to increase the expression of filaggrin, caspase-14, and claudin-1, which may facilitate the production of natural moisturizing factors and strengthen the tight junction structure [5,7,8]. It also potentiates the anti-inflammaging system in epidermal keratinocytes [9]. In parallel with this, clinical studies have revealed that the daily application of Pitera^TM^ indeed increases the skin hydration and decreases the TEWL of facial skin [10,11]. It also alleviates or completely restores mask-induced skin damage [11].

In the present study, we aimed to evaluate whether the daily application of GFF-containing skin care products can modify the long-term aging process. We measured the signs of facial skin aging (wrinkles, hyperpigmented spots, roughness, skin hydration, TEWL, and elasticity) in 1999 and 2010 in 86 Japanese women. Then, all subjects applied three GFF-containing skin care products (G3 products; SK-II Facial Treatment Essence, SK-II Cellumination Essence, and SK-II Skin Signature Cream) twice daily for 12 months (M), with these skin parameters being measured at 2 M, 8 M, and 12 M during this period. There was significant aggravation of facial aging parameters such as wrinkles, spots, and roughness in all 86 participants during the 11-year interval. This aging process was associated with reduced hydration and increased TEWL. Notably, twice-daily application of G3 products significantly and cumulatively increased skin hydration with a correlated reduction of TEWL during the 12 M treatment period. Such treatment also significantly and cumulatively reversed the 11-year facial aging in terms of three parameters: wrinkles, spots, and roughness. These results suggest that facial skin retains the potential to recover from the aging process when appropriate cosmetic agents are applied to it.

## 2. Materials and Methods

### 2.1. Subjects and Study Protocol

The skin evaluation was performed in 1999 and 2010 (11 years later) on 86 healthy Japanese females who either worked indoors or were housewives and lived in Akita City, Japan. The examination room was maintained at a constant temperature and humidity (room temperature 20 ± 2 °C, relative humidity 50 ± 5%). To avoid any influence of seasonal variations, both the 1999 and 2010 studies were performed in late March. Immediately after the skin measurement in 2010 (0 M), all 86 participants started to apply the G3 products twice daily for 12 M. The facial skin parameters were measured at 2 M, 8 M, and 12 M during this period.

The age of the subjects ranged from 5 to 64 years old [mean ± standard deviation (SD), 38.1 ± 16.9] in 1999, and from 16 to 75 years old (49.1 ± 16.9) in 2010. The number of subjects by age group in 2010 was as follows: 8 in their teens (17.5 ± 1.20), 7 in their 20s (23.7 ± 3.09), 7 in their 30s (38.0 ± 1.29), 20 in their 40s (44.5 ± 2.32), 19 in their 50s (55.9 ± 3.08), 15 in their 60s (65.1 ± 3.18), and 10 in their 70s (71.2 ± 1.40). None of the subjects underwent any type of esthetic treatment such as laser cosmetic procedures during the study period. The study protocol was approved by the Ethical Committee of Global Product Stewardship in P&G Innovation Godo Kaisya (ethical approval number CT10-016). Written informed consent was obtained from all subjects prior to enrollment in the study.

### 2.2. Facial Optical Imaging and Objective Image Analysis

The subjects washed their faces using the prescribed cleansing foam and then spent 20 min becoming accustomed to the environment of the measurement room at a constant temperature and humidity. Each subject’s face was photographed using an image capture system (Beauty Imaging System; BIS1, P&G, : Cincinnati, OH, USA) (Figure 1) consisting of a high-resolution digital camera with a close-up lens (Fujifilm DS330, Tokyo, Japan), fluorescent lighting (5500 K, EFS13UED, Panasonic Corporation, Osaka Japan), and an anchor to fix the jaw and forehead to take the photos in the same position, as reported previously [1]. The camera was calibrated for each measurement, which made it possible to apply the same measurement conditions, including the position of the face, for both the 1999 and 2010 measurements. The region of interest (ROI) of the images was from the outer edge of the eyes to the cheek, and the following characteristic objects were extracted by measuring the contrast in the shape and pixels using an image analysis algorithm [1]. Wrinkles were defined as ≥ 5 mm in length, perimeter/length ratio ≤ 2.5, and circularity (perimeter^2^/area) ≥ 34. Total wrinkle area fraction was quantified as follows: total wrinkle area (pixels)/ROI (pixels). Hyperpigmented spots were defined as ≥ 5 mm^2^ in area, color contrast DeltaE ≥ 3 compared with the surrounding skin region, and circularity (perimeter^2^/area) ≥ 20. Total hyperpigmented area fraction was also quantified as follows: total hyperpigmented area (pixels)/ROI (pixels). As an index of skin surface roughness, total texture area fraction [total texture area (pixels)/ROI (pixels)] was quantified as ≤ 3 mm^2^ in area, aspect ratio ranging from 0.5 to 2, and color contrast DeltaE ≥ 1.5, while pores were defined as ≤ 4 mm^2^ in area, color contrast DeltaE ≥ 2, and circularity (perimeter^2^/area) ≥ 20, which differ from hyperpigmented spots in terms of size and circularity. Total pore area fraction was quantified as follows: [total pore area (pixels)/ROI (pixels)] × 3. Facial skin color (lightness, redness, yellowness) was also measured in the region of interest. The mean values of the resulting data obtained by these evaluations were analyzed. In accordance with the above-described procedures, the wrinkle, spot, and roughness parameters were fitted on a scale from 0 to 1, and the average scale scores of wrinkles, spots, and roughness were summated and defined as Overall Skin Aging Score. A higher Overall Skin Aging Score indicated greater skin aging. Based on the Overall Skin Aging Score, reversal years of skin aging compared to own 11-year longitudinal skin aging was estimated as follows:$$\mathrm{Reveal}\;\mathrm{Years}\;\mathrm{of}\;\mathrm{Skin}\;\mathrm{Aging}=\left(\frac{\Delta \;\mathrm{Overall}\;\mathrm{Aging}\;\mathrm{Score}\;\mathrm{from}\;2M,8M\;\mathrm{or}\;12M-1999\;}{\Delta \;\mathrm{Overal}\;\mathrm{Aging}\;\mathrm{Score}\;\mathrm{from}\;2010-1999}\right)\;x\;11\left(\mathrm{years}\right)$$

### 2.3. Biophysical Measurements

Skin hydration (water content), TEWL, and skin elasticity of the cheek were measured using Corneometer1 (Courage + Khazaka Electronic GmbH, Cologne, Germany), Tewameter^TM^ (Courage + Khazaka), and Cutometer1 (Courage + Khazaka), respectively. Sebum excretion at the forehead was measured with a Sebumeter (Courage + Khazaka) 1 h after cleaning of the face [1].

### 2.4. Statistical Analysis

Pearson’s correlation coefficients (r) between all seven skin optical parameters [image analysis data on wrinkles, hyperpigmented spots, texture, pores, and skin color (lightness, redness, yellowness)] and age were examined in both the 1999 and 2010 studies. Quantitative comparison of those skin parameters from 1999 and 2010 (0 M, 2 M, 8 M, and 12 M) was also performed by age group using two-way ANOVA (significance level *p* < 0.05).

## 3. Results

### 3.1. Aggravation of Skin Aging Parameters from 1999 to 2010

We measured various facial skin parameters in 1999 and 2010 in 86 female volunteers. As expected, the facial skin hydration was significantly decreased with a reciprocal increase in TEWL in 2010 compared with the findings in 1999 (Table 1). In parallel with this, three aging-related parameters, namely wrinkles, hyperpigmented spots, and roughness, were also significantly aggravated during the 11-year interval (Table 1). In addition, facial skin yellowness was increased, while its lightness was decreased in 2010 compared with the findings in 1999 (Table 1). Facial redness was not significantly altered between 1999 and 2010. Facial pore size and sebum production were also significantly increased in the aging process (Table 1). Elasticity was significantly decreased in 2010 compared with the level in 1999 (Table 1).

### 3.2. Application of G3 Products Reversed Facial Skin Aging

To determine whether the application of G3 products can reverse facial skin aging, twice-daily application of such products was performed by the 86 subjects for 1 year in 2010 after the initial skin measurement [2010 (0 M)]. The facial skin parameters were also measured at 2 M, 8 M, and 12 M. Chronological facial skin measurements (mean (S.D.)) in 1999 and 2010 (at 0, 2, 8, and 12 M) are shown in Appendix A.

As expected, the treatment with G3 products significantly increased the skin hydration, even at 2 M (Figure 2A). It was further increased to the 1999 baseline levels at 8 M and 12 M (Figure 2A). Meanwhile, the elevated TEWL values at 0 M in 2010 were gradually and significantly decreased and returned to the 1999 baseline levels after the 12 M treatment with the G3 products (Figure 2B).

Notably, the aggravated wrinkles (Figure 3A), spots (Figure 3B), and roughness (Figure 4A) at 0 M in 2010 were also gradually and significantly decreased to the 1999 baseline levels by the application of G3 products during the 12 M treatment period. A previous study revealed that the facial aged skin appearance was well correlated with Overall Skin Aging Score calculated from the values of wrinkles, spots, and roughness [1]. The Overall Skin Aging Score was significantly elevated from 1999 to 2010 (0 M), but it was recovered to a level close to that in 1999 by treatment with G3 products (Figure 4B).

Based on the Overall Skin Aging Score, we then estimated the number of reversed years from the actual 11 years of aging (Figure 5). The daily application of G3 products clearly rejuvenated the facial skin appearance. The estimated numbers of reversed years were −4.77 ± 1.24 years at 2 M, −7.07 ± 2.58 years at 8 M, and −9.23 ± 3.19 years at 12 M (Figure 5A). We next examined the relationship between change of Overall Skin Aging Score (2000 0 M—1999) and change of Overall Skin Aging Score (2000 12 M–0 M). There was a negative correlation (*p* < 0.01) between change of Overall Skin Aging Score of 11 years (0M—1999) and change of overall skin aging score (12 M–0 M) (Figure 5B). This indicated that greater overall skin aging over 11 years was associated with greater reversal of the overall skin aging by using the G3 products for 12 M. Chronological facial images of three subjects are shown in Figure 6, Figure 7 and Figure 8. 

Notably, subgroup analysis revealed that G3 products exerted the potent antiaging effects against wrinkles, spots, and roughness similarly in subjects of different age groups (30s through 50s or more) (Appendix A).

### 3.3. Change of Skin Hydration Was Negatively Correlated with Change of Skin Aging Parameters

We next examined whether change of skin hydration (12 M–0 M) is associated with skin aging parameters. The net values of each parameter were also summarized in Appendix A. As expected, a significant negative correlation was observed between change of skin hydration (12 M–0 M) and change of skin TEWL (12 M–0 M) (r = −0.322, *p* = 0.028, Figure 9A). Change of skin hydration (12 M–0 M) was also negatively correlated with change of wrinkles (12 M–0 M) (r = −0.373, *p* = 0.025, Figure 9B), change of spots (12 M–0 M) (r = −0.430, *p* = 0.018, Figure 10A), and change of roughness (12 M–0 M) (r = −466, *p* < 0.01, Figure 10B).

### 3.4. Effects of G3 Product Application on Other Skin Parameters

We next analyzed whether treatment with G3 products affects other facial parameters. Facial pore size (Figure 11A) and sebum production (Figure 11B) were increased during the aging process from 1999 to 2010 (0 M). Treatment with G3 products significantly reduced both parameters in 2010 (12 M) to levels close to those in 1999.

Facial skin lightness (Figure 12A) was deteriorated (became darker) during the aging process but was also recovered to the 1999 level by the application of G3 products. In contrast, facial skin yellowness was significantly increased in 2010 (0 M) compared with the level in 1999. Daily application of G3 products gradually decreased the facial skin yellowness (Figure 11B). Significant reduction of yellowness was observed after 12 M application of G3 products in 2010 (12 M) (Figure 12B).

In contrast, facial skin redness was not affected by the decade-long aging process, but the daily application of G3 products significantly decreased the redness after 12 M application (Figure 13A). Facial skin elasticity was significantly reduced in 2010 (0 M) compared with that in 1999 (Figure 13B), which was reversed to the 1999 level by 12 M application of G3 products (Figure 13B).

## 4. Discussion

Aging is a natural fate of all living organisms. However, facial skin aging is a major concern of humans, especially women. Physiological or chronological aging of the skin is accelerated by various external stresses such as ultraviolet (UV) radiation, environmental pollutants, and mechanical stress [12,13]. These exogenous stresses stimulate keratinocytes to produce proinflammatory cyto/chemokines [14,15,16,17,18], which may facilitate the aging process referred to as inflammaging [12,13]. Over the course of natural aging, facial skin gradually acquires signs of aging including wrinkles, hyperpigmented spots, roughness, pore enlargement, and decreased elasticity [1,2,19,20,21].

In the present study, we clearly demonstrated that 11 years of aging did aggravate parameters representing facial skin aging, namely wrinkles, spots, and roughness, in 86 female subjects who participated in 1999 and 2010. These three parameters are known to be particularly important for recognizing facial aging in females visually [1]. The Overall Skin Aging Score defined based on the values of wrinkles, spots, and roughness well differentiates younger-looking faces from older-looking ones [1]. The Overall Skin Aging Score was also significantly elevated in 2010 compared with the level in 1999 in this study.

Notably, the twice-daily application of G3 products for 12 M successfully reduced the decade-long increments of wrinkles, spots, and roughness. Significant reductions of these parameters were detected as early as 2 M after the initiation of application, and the reduction rates increased until the end of 12 M. The estimated mean number of reversed years was −9.2. Considering the actual aging period (11 years), it was highly satisfactory that the rejuvenating effect on the participating women’s skin led it to appear over 9 years younger after 1 year of using G3 products.

In addition to wrinkles, spots, and roughness, other parameters such as pore size, sebum production, and elasticity were significantly affected by the 11 years of aging. The treatment with G3 products also significantly decreased or reversed the enlarged pore size and increased sebum production associated with the decade-long aging process. The aging-induced decrease of facial skin elasticity was also significantly rescued by the application of G3 products. As has been reported previously [10,11], G3 products were also capable of ameliorating the facial skin redness in this study. However, redness itself was not a suitable parameter to represent the facial skin aging because the redness values stayed unchanged from 1999 to 2010 at least in the present Chinese female volunteers.

G3 products have been proved to increase skin hydration and decrease skin TEWL in clinical studies [10,11]. In the present study, a significant increase of hydration was confirmed as early as 2 M after the initiation of G3 product use and it was further increased at 8 M and 12 M by continuous application. The increased hydration may explain, at least in part, why the G3 products exhibit antiaging effects because the increased hydration was significantly correlated with antiwrinkle, antispot, and antiroughness effects of G3 products. GFF is known to activate aryl hydrocarbon receptor and enhance the expression of filaggrin, which is an essential source of natural moisturizing factors [5,9]. GFF also accelerates the production of the anti-inflammatory cytokine IL-37 in epidermal keratinocytes and contributes to anti-inflammaging [9,22]. Moreover, GFF is capable of inhibiting the expression of cyclin-dependent kinase inhibitor 2A (CDKN2A or p16INK4A), which is a critical biomarker for facial senescence in epidermal keratinocytes [6,23]. We cannot rule out the possibility that these biological functions of GFF contribute to the potential of G3 products to reverse the skin aging process.

Limitations of this study include that we did not address how well the antiaging effects of G3 products were sustained after the cessation of their use. We also do not know the mechanism by which the topical application of G3 products increased the skin elasticity. In addition, we did not know yet whether the redness values are affected by facial skin aging process in Caucasian population. Some studies revealed that niacinamide or retinol decreased the wrinkle and pigmented spots [24,25]. However, there has been no decade-long longitudinal studies to evaluate the rejuvenating potential of certain cosmeceutical products. The present 11-year longitudinal study was the first one to prove the reversing potential against facial skin aging by skin care products.

In conclusion, this study showed that the parameters of facial skin aging of wrinkles, spots, and roughness were indeed aggravated over an 11-year interval in 86 female volunteers. Twice-daily application of G3 products significantly reversed the deterioration of aged skin parameters and rejuvenated the facial skin in terms of its appearance. These results suggest that aged facial skin may be rejuvenated by applying appropriate cosmetic agents.

## Figures and Tables

**Figure 1 jcm-12-01168-f001:**
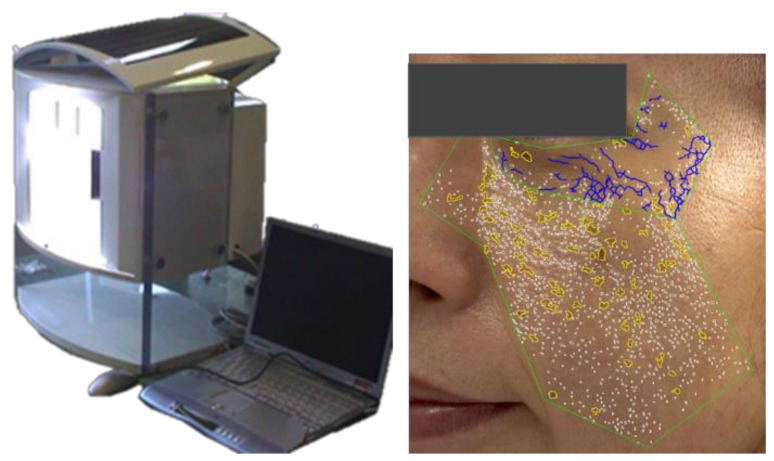
Beauty Imaging System (BIS) and image analysis overlays of wrinkles, spots, roughness, and pores.

**Figure 2 jcm-12-01168-f002:**
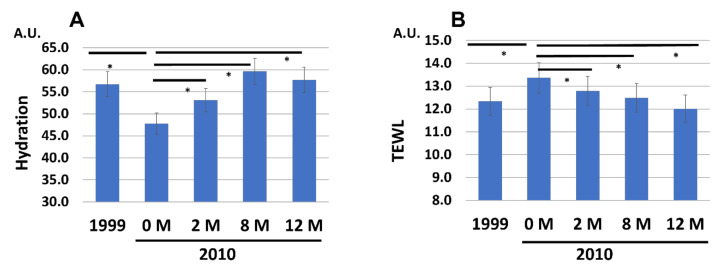
Chronological facial skin measurements in 1999 and in 2010 (at 0, 2, 8, and 12 M). All 86 subjects were treated with twice-daily application of G3 products for 12 M (0 M to 12 M). (**A**) Skin hydration. (**B**) TEWL. * *p* < 0.05.

**Figure 3 jcm-12-01168-f003:**
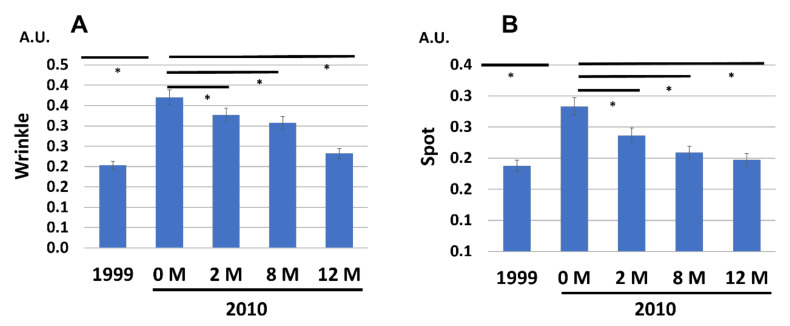
Chronological facial skin measurements in 1999 and in 2010 (at 0, 2, 8, and 12 M). All 86 subjects were treated with twice-daily application of G3 products for 12 M (0 M to 12 M). (**A**) Wrinkles. (**B**) Hyperpigmented spots. * *p* < 0.05.

**Figure 4 jcm-12-01168-f004:**
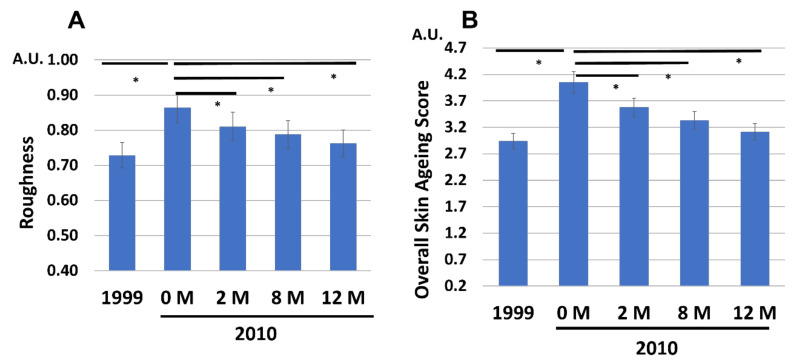
Chronological facial skin measurements in 1999 and in 2010 (at 0, 2, 8, and 12 M). All 86 subjects were treated with twice-daily application of G3 products for 12 M (0 M to 12 M). (**A**) Roughness. (**B**) Overall Skin Aging Score. * *p* < 0.05.

**Figure 5 jcm-12-01168-f005:**
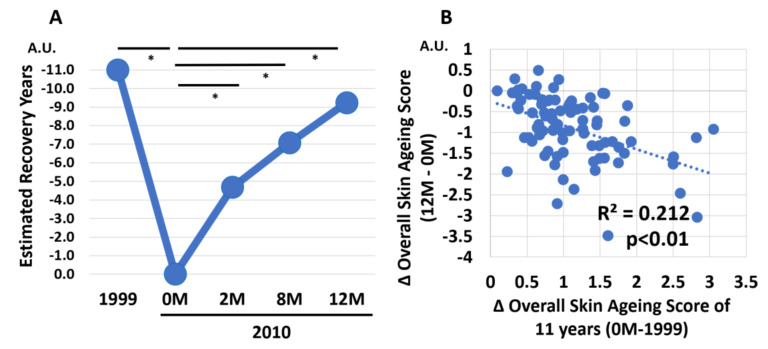
(**A**) Estimated number of recovered years in 1999 and in 2010 (at 0, 2, 8, and 12 M). All 86 subjects were treated with twice-daily application of G3 products for 12 M (0 M to 12 M). * *p* < 0.05. (**B**) Negative correlation (*p* < 0.01) between change of Overall Skin Aging Score of 11 years (0 M—1999) and change of Overall Skin Aging Score (12 M–0 M) after twice-daily application of G3 products for 12 M.

**Figure 6 jcm-12-01168-f006:**
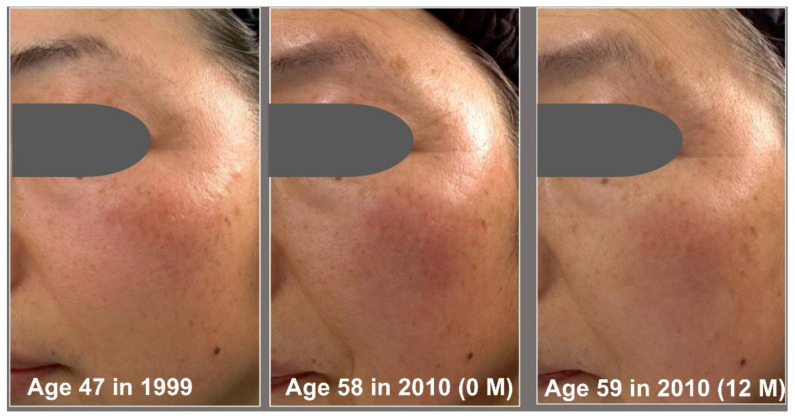
Chronological facial images of Case 1 (age: 47 in 1999). Twice-daily application of G3 products was performed from 2010 (0 M) to 2010 (12 M). The visual facial appearance aged from 1999 to 2010 (0 M), but it was rejuvenated in 2010 (12 M) to close to the level in 1999.

**Figure 7 jcm-12-01168-f007:**
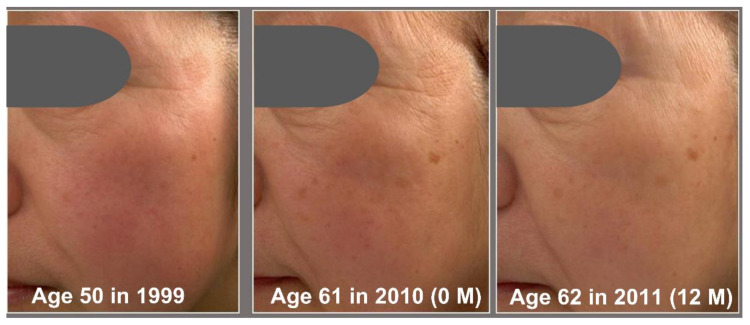
Chronological facial images of Case 2 (age: 50 in 1999). Twice-daily application of G3 products was performed from 2010 (0 M) to 2010 (12 M). Visual facial appearance aged from 1999 to 2010 (0 M), but it was rejuvenated in 2010 (12 M) to close to the level in 1999.

**Figure 8 jcm-12-01168-f008:**
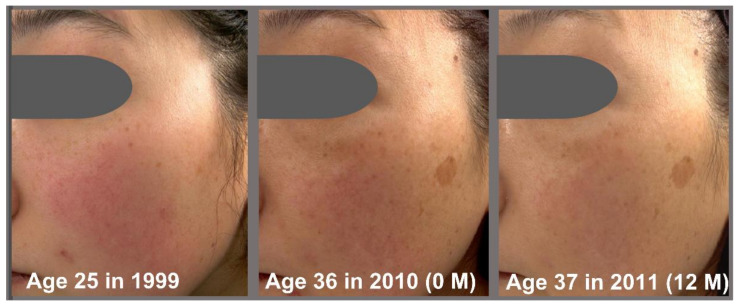
Chronological facial images of Case 3 (age: 25 in 1999). Twice-daily application of G3 products was performed from 2010 (0 M) to 2010 (12 M). Visual facial appearance aged from 1999 to 2010 (0 M), but it was rejuvenated in 2010 (12 M) to close to the level in 1999.

**Figure 9 jcm-12-01168-f009:**
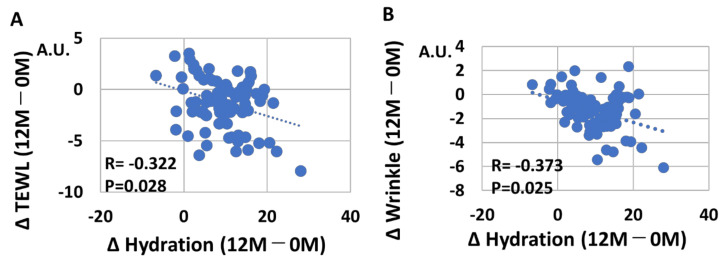
(**A**) Negative correlation (*p* = 0.028) between Δ skin hydration (12 M–0 M) and Δ skin TEWL (12 M–0 M). (**B**) Negative correlation (p = 0.025) between Δ skin hydration (12 M–0 M) and Δ wrinkles (12 M–0 M).

**Figure 10 jcm-12-01168-f010:**
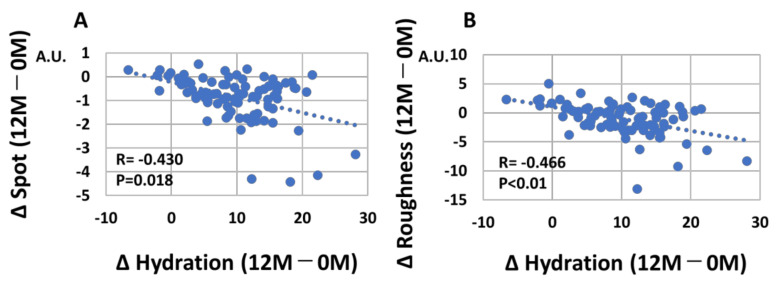
(**A**) Negative correlation (*p* = 0.018) between Δ skin hydration (12M–0 M) and Δ spots (12 M–0 M). (**B**) Negative correlation (*p* < 0.01) between Δ skin hydration (12 M–0 M) and Δ wrinkles (12 M–0 M).

**Figure 11 jcm-12-01168-f011:**
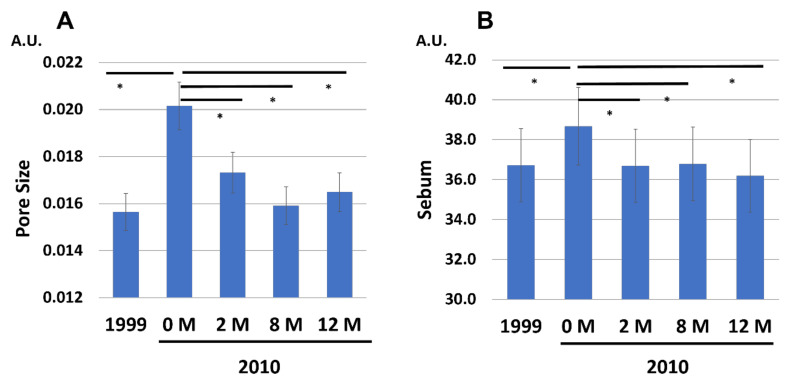
Chronological facial skin measurements in 1999 and in 2010 (at 0, 2, 8, and 12 M). All 86 subjects were treated with twice-daily application of G3 product formula for 12 M (0 M to 12 M). (**A**) Pore size. (**B**) Sebum production. * *p* < 0.05.

**Figure 12 jcm-12-01168-f012:**
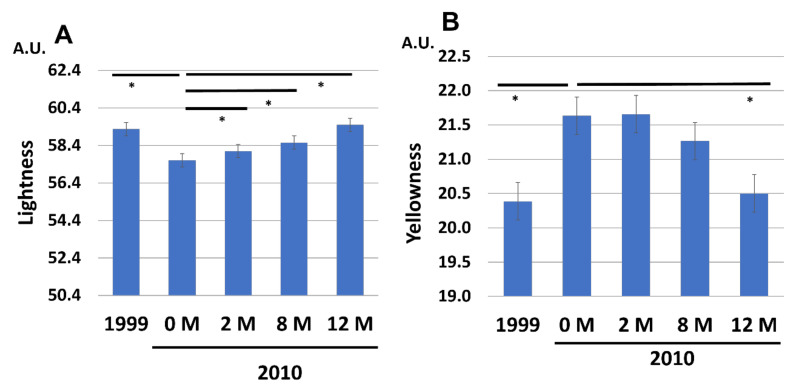
Chronological facial skin measurements in 1999 and in 2010 (at 0, 2, 8, and 12 M). All 86 subjects were treated with twice-daily application of G3 products for 12 M (0 M to 12 M). (**A**) Skin lightness. (**B**) Skin yellowness. * *p* < 0.05.

**Figure 13 jcm-12-01168-f013:**
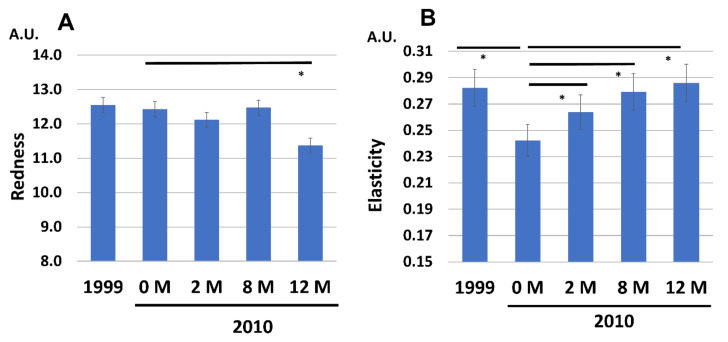
Chronological facial skin measurements in 1999 and in 2010 (at 0, 2, 8, and 12 M). All 86 subjects were treated with twice-daily application of G3 products for 12 M (0 M to 12 M). (**A**) Skin redness. (**B**) Skin elasticity. * *p* < 0.05.

**Table 1 jcm-12-01168-t001:** Changes in skin aging parameters between 1999 and 2010 (0 M).

	1999	2010 (0 M)	*p*-Value
Hydration	56.71 ± 7.11	47.76 ± 8.68	<0.01
TEWL	12.34 ± 3.83	13.36 ± 3.18	<0.01
Wrinkles	0.2 ± 0.18	0.37 ± 0.28	<0.01
Spots	0.19 ± 0.11	0.28 ± 0.15	<0.01
Roughness	0.73 ± 0.4	0.86 ± 0.42	<0.01
Lightness	59.27 ± 2.69	57.6 ± 2.45	<0.01
Redness	12.55 ± 1.92	12.43 ± 1.54	0.352
Yellowness	20.39 ± 1.93	21.63 ± 1.93	0.028
Pore size	0.0157 ± 0.0122	0.0202 ± 0.011	<0.01
Sebum	36.73 ± 29.94	38.68 ± 34.11	0.033
Elasticity	0.28 ± 0.05	0.24 ± 0.06	<0.01

## Data Availability

The data presented in this study are available on request from the corresponding author. The data are not publicly available because of privacy restrictions.

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
