# Peer review of "Significant Reversal of Facial Wrinkle, Pigmented Spot and Roughness by Daily Application of Galactomyces Ferment Filtrate-Containing Skin Products for 12 Months—An 11-Year Longitudinal Skin Aging Rejuvenation Study"

_jcm, 2023, doi:10.3390/jcm12031168_

Round 1

Reviewer 1 Report

The authors presented an interesting original article for the use of G3 products in facial skin aging. Despite, significant findings, in my view, there are some important issues needed to be addressed in order the manuscript proceed to publication.

-       “Introduction”: objectives should be defined, instead of results in that section

-       “Methods”: “reversed years” should be clearly described in methods and how they are calculated. All others parameters are clearly described on methods.

-       “Results”: First, when the correlation among entities is referred, they authors might also present the value of pearson correlation, in order to make clear to the reader if there is a strong, moderate or weak correlation. Also,  for the presentation of change in scores, a table with mean (SD) change of scores along with referral of the change, except for statistical significance. So, the authors might present the value of change in scores along with p-value as a table and in the main text. For example, statements in s232-234 are very general, without proving the change of each parameter.

In concordance with, some graphs could be presented as supplementary material, because they render the manuscript incomprehensible.

Also, in redness, despite the fact that between 1999 and 2010 there was not any change in redness, after application of G3, there was a statistically significant difference. How the authors explain that?

-       “Discussion”: that section needs major revision, as it should be more comprehensive and critical to final results. For example, discussing findings  from studies for other similar products, might make clearer the benefit of the use of G3 products. Another point to investigate is that if any differentiation in skin aging parameters per age group after treatment was noticed. Is the same result for all age groups in the sample?

Overall, a well-structured and interesting study which needs major improvement in order to proceed to publication.

Author Response

Reply to the Reviewer 1

The authors presented an interesting original article for the use of G3 products in facial skin aging. Despite, significant findings, in my view, there are some important issues needed to be addressed in order the manuscript proceed to publication.

→ Thank you very much for your encouraging comment.

“Introduction”: objectives should be defined, instead of results in that section

→ Thank you very much for your helpful comment. According to your comment, we added the objectives in the Introduction as underlined.

“In the present study, we aimed to evaluate whether the daily application of GFF-containing skin care products can modify the long-term aging process.”

“Methods”: “reversed years” should be clearly described in methods and how they are calculated. All others parameters are clearly described on methods.

→ Thank you very much for your critical comment. According to your comment, we added the calculation method in the Methods as underlined.

“In accordance with the above-described procedures, the wrinkle, spot, and roughness parameters were fitted on a scale from 0 to 1, and the average scale scores of wrinkles, spots, and roughness were summated and defined as Overall Skin Aging Score. A higher Overall Skin Aging Score indicated greater skin aging. Based on the Overall Skin Aging Score, reversal years of skin aging compared to own 11-year longitudinal skin aging was estimated as follows:

 “

“Results”: First, when the correlation among entities is referred, they authors might also present the value of pearson correlation, in order to make clear to the reader if there is a strong, moderate or weak correlation.

→ Thank you very much for your helpful comment. According to your comment, we added the values of pearson correlation in Results as underlined.

“ change of skin TEWL (12 M - 0 M) (r = - 0.322, P = 0.028, Figure 9A). Change of skin hydration (12 M - 0 M) was also negatively correlated with change of wrinkles (12 M - 0 M) (r = - 0.373, P = 0.025, Figure 9B), change of spots (12 M - 0 M) (r = - 0.430, P = 0.018, Figure 10A), and change of roughness (12 M - 0 M) (r = - 466, P < 0.01, Figure 10B).”

Also, for the presentation of change in scores, a table with mean (SD) change of scores along with referral of the change, except for statistical significance. So, the authors might present the value of change in scores along with p-value as a table and in the main text. For example, statements in s232-234 are very general, without proving the change of each parameter.

→ Thank you very much for your important comment. According to your comment, we summarized the chronological net values of each parameter in Supplementary Table S1.

“We next examined whether change of skin hydration (12 M - 0 M) is associated with skin aging parameters. The net values of each parameter were also summarized in Supplementary Table S1. As expected, a significant negative correlation was observed between change of skin hydration (12 M - 0 M) and change of skin TEWL (12 M - 0 M) (r = - 0.322, P = 0.028, Figure 9A). Change of skin hydration (12 M - 0 M) was also negatively correlated with change of wrinkles (12 M - 0 M) (r = - 0.373, P = 0.025, Figure 9B), change of spots (12 M - 0 M) (r = - 0.430, P = 0.018, Figure 10A), and change of roughness (12 M - 0 M) (r = - 466, P < 0.01, Figure 10B).”

In concordance with, some graphs could be presented as supplementary material, because they render the manuscript incomprehensible.

→ Thank you very much for your helpful comment. According to your comment, we presented our data in Supplementary Table S1 and Supplementary Figure S1.

Also, in redness, despite the fact that between 1999 and 2010 there was not any change in redness, after application of G3, there was a statistically significant difference. How the authors explain that?

→ Thank you very much for your helpful comment. According to your comment, we added the following sentences in Discussion as underlined.

“As has been reported previously [10,11], G3 products were also capable of ameliorating the facial skin redness in this study. However, redness itself was not a suitable parameter to represent the facial skin aging because the redness values stayed unchanged from 1999 to 2010 at least in the present Chinese female volunteers.”

“In addition, we did not know yet whether the redness values are affected by facial skin aging process in Caucasian population.”

“Discussion”: that section needs major revision, as it should be more comprehensive and critical to final results. For example, discussing findings from studies for other similar products, might make clearer the benefit of the use of G3 products.

→ Thank you very much for your critical comment. Some studies (ref #24 and ref#25) revealed that niacinamide or retinol decreased the wrinkle and pigmented spots. However, there has been no decade-long longitudinal study to evaluate the rejuvenation potential of certain cosmeceutical products. Our study was the first one to evaluate the reversing potential against facial skin aging by skin care products. According to your comment, we added the following sentences in the Discussion as underlined.

“In addition, we did not know yet whether the redness values are affected by facial skin aging process in Caucasian population. Some studies revealed that niacinamide or retinol decreased the wrinkle and pigmented spots [24,25]. However, there has been no decade-long longitudinal studies to evaluate the rejuvenating potential of certain cosmeceutical products. The present 11-year longitudinal study was the first one to prove the reversing potential against facial skin aging by skin care products.”

Another point to investigate is that if any differentiation in skin aging parameters per age group after treatment was noticed. Is the same result for all age groups in the sample?

→ Thank you very much for your helpful comment. According to your comment, we performed a subgroup analysis. We added the following sentences in the Results as underlined.

“Notably, subgroup analysis revealed that G3 products exerted the potent anti-aging effects against wrinkles, spots and roughness similarly in subjects of different age groups (30s through 50s or more) (Supplementary Figure S1).”

Overall, a well-structured and interesting study which needs major improvement in order to proceed to publication.

→ Thank you very much again for critical comments. According to your comments, we revised the article. We hope that the revised article is now suitable for publication in JCM.

Reviewer 2 Report

This is a well-designed clinical trial in which objective parameters of clinical outcome were analyzed.
The title needs improvement and I suggest focusing on the evaluated outcomes.

-The research investigated the clinical response of Galactomyces fermente filtrate-containing skin care products applied in face twice daily for 12 months
-This study is very interesting and can contribute to answer a lacuna about the clinical improvement promoted by topical agents.
--It adds information about a principle less studied in other researchs.
-No consideration about methodology.
-The references are appropriate.
-The authors should rewrite the title, trying to reflect the  outcomes analysed (instead of including a type of mathematics analysis that resulted in an estimation of rejuvenastion).

Author Response

Reply to the Reviewer 2

This is a well-designed clinical trial in which objective parameters of clinical outcome were analyzed.

→ Thank you very much for your encouraging comment.

The title needs improvement and I suggest focusing on the evaluated outcomes.

→ Thank you very much for your helpful comment. According to your comment, we amended the title as follows.

“Significant reversal of facial wrinkle, pigmented spot and roughness by daily application of Galactomyces ferment filtrate-containing skin products for 12 months - An 11-year longitudinal facial skin aging rejuvenation study”

Thank you very much again for critical comments. According to your comments, we revised the article. We hope that the revised article is now suitable for publication in JCM.

Round 2

Reviewer 1 Report

The manuscript is significantly improved after appropriate changes.

I have no more comments for the text!